# COVID-19, government measures and hospitality industry performance

David Yechiam Aharon[1]*, Arie Jacobi[1], Eli Cohen[2], Joseph Tzur[3], Mahmoud Qadan[4]

**1** Department of Business Administration, Ono Academic College, Kiryat Ono, Israel, **2** Department of Management, Guilford Glazer Faculty of Business and Management, Ben-Gurion University of the Negev, Beer Sheva, Israel, **3** Department of Economics and Accounting, Ruppin Academic Center, Emek Hefer, Israel, **4** Department of Business Administration, Faculty of Management, University of Haifa, Haifa, Israel

* dudi.ah@ono.ac.il

**Data Availability Statement:** All relevant data are within the manuscript and its Supporting Information files.

**Funding:** The author(s) received no specific funding for this work.

## Abstract

This study explores the interplay between public measures adopted by the U.S. government to combat COVID-19 and the performance of the American hospitality industry. The recent global pandemic is a natural experiment for exploring the role of government interventions and their direct impact on hospitality stock returns in the U.S. financial market. Overall, our findings show that most of the government interventions were associated with a negative response in the returns of the hospitality industry, a response that became more negative as the COVID-19 pandemic evolved. Similar patterns were also detected for other industries such as entertainment and transportation that are closely related to hospitality. The findings we document are fundamental to understanding the trends and fluctuations in hospitality stocks in the current crisis and any similar crisis in the future.

## Introduction

The recent COVID-19 crisis may be one of the most influential and unprecedented events for firms, investors, policy makers and many other market participants. Along with the worldwide outbreak of the disease, it has also spilled over economically to major capital markets and sectors, thereby also adversely affecting the performance and stability of the hospitality industry.

The negative impact of the COVID-19 crisis is mainly affecting service-oriented sectors such as the hospitality industry. The latter functions as a powerful vehicle for economic growth and job creation all over the world. It is directly and indirectly responsible for regional development, numerous types of jobs, industries and sub-industries, and underpin many economic activities. According to the U.S. travel association (https://www.ustravel.org/research/travel-facts-and-figures), in 2019, travel alone generated $2.6 trillion in total economic output, supported 15.8 million American jobs, and accounted for 2.9% of the U.S. gross domestic product (GDP). These statistics highlight the economic importance of travel and tourism to the U.S. economy as well as to the global economy as a whole.

According to the World Health Organization (WHO), the COVID-19 pandemic was first reported in Wuhan, China on December 31, 2019. The pandemic spread quickly all over Asia, leaving behind it health and economic crises. On March 2, 2020, COVID-19 was first reported

**Competing interests:** The authors have declared
that no competing interests exist.

in the US and 10 days later Europe became the epicenter of the pandemic, both leading to even worse health and economic catastrophes. On June 1, 2020, there were over 6 million confirmed cases, and more than 370,000 confirmed deaths worldwide. Remarkably, as of May 20, 2021, the WHO reported that there had been 166,346,635 confirmed cases of COVID-19, including 3,449,117 deaths.

During the COVID-19 crisis, governments have taken different measures in the health, public and economic fields. These interventions were aimed at containing the spread of the virus in an attempt to minimize the adverse effects of the COVID-19 outbreak on both the health and economic realms. A brief review of such interventions reveals that governments imposed different actions such as canceling public gatherings, closing workplaces and schools, requiring social distancing, and also providing economic support, creating contact tracing and offering COVID-19 testing policies.

Using a unique dataset by Hale et al. [1] tracking the U.S. government's interventions to the COVID-19 spread, we explore the response of U.S. hospitality stocks to different types of government interventions. In addition, we extend our examinations to additional firms operating in sectors closely related to the hospitality industry, given the possibility of a spillover effect from the hospitality industry to its close sub-industries.

We contribute to the literature in three areas. First, we add to the growing body of research dealing with the impact of epidemics and crises on asset pricing (e.g., [2–15]) by examining the immediate and short-term effect of the COVID-19 outbreak on the price evolution of stocks in the hospitality sector using the event study method. Second, we contribute to the literature dealing with the impact of government interventions and their reflection in asset prices during times of crisis (e.g., [11, 16–22]) by sorting the intervention into economic (e.g., income support, debt contract relief) and non-economic (e.g., travel restrictions, school closings) measures and exploring their consequences. Overall, the results show that both economic and non-economic interventions imposed on the public can affect the prices of hospitality stocks. The magnitude of the short-term negative effect increases with the timeline of the evolution of the epidemic and the increased level of uncertainty.

Finally, we contribute to the literature dealing with changes in government policy and uncertainty in the stock market (e.g., [23]), by exploring the Granger-causality relationship between uncertainty due to infectious disease and variations in the hospitality stock returns. In addition, we examine how uncertainty reacts to government intervention. The results highlight that the hospitality business is very sensitive to economic uncertainty. When faced with adverse economic conditions, consumers typically tend to postpone using disposable income for travel and tourism in favor of more basic needs [24].

Our main empirical findings documented here show that the major challenge during the spread of COVID-19 was uncertainty. This uncertainty originated in two different, yet related, sources. The first source stemmed from the pandemic itself and the intensified ambiguity about the real consequences for the economy in terms of the time required for economic recovery, the rapidity of the spread of the infection and its lethality. This contention is confirmed using textual analysis of unique data from approximately 3,000 U.S. newspapers.

The second source was related to the uncertainty originating in the government interventions themselves. There was a great deal of ambiguity about the economic and non-economic consequences of government interventions. In addition, the public was uncertain about whether the government planned future interventions.

In this spirit, empirical studies have shown that increased ambiguity about government policy and spending has direct implications for the steady state of many macroeconomic variables such as debt, the GDP and consumption (e.g., [25]).

To summarize, during a crisis like a pandemic, the leading challenge a government faces is reducing both types of uncertainty. Doing so is vital for industries that are sensitive to such uncertainty such as the hospitality sector. Policy makers might be well-advised to impose measures with detailed transparency about their long-term plans to promote certainty. Ambiguity about current and future government spending (and stimulus packages) creates uncertainty in the stock market [23] and disrupts many macroeconomic variables such as debt, GDP and consumption (e.g., [25]). Our findings can also help policy makers fine-tune their aid policy and help tourism planners prepare better for possible future government interventions during subsequent epidemics such as COVID-19.

The remainder of this study is organized as follows. Section 2 presents the scientific background. Section 3 describes the data sources, our research methodology, and the measurement of the variables. Section 4 details the empirical findings, Section 5 discusses policy implications and future recommendations, while the last section provides a summary of the findings.

## Literature review

### The effects of government interventions during COVID-19

Zaremba et al. [22] examined the impact of government interventions on stock market liquidity in 49 countries during January-April 2020, and demonstrated that the effect of government interventions is limited in scale and scope. Specifically, they reported that the closures of workplaces and schools reduced liquidity levels in emerging markets, while COVID-19 information campaigns promoted trading activity. In a subsequent study, Zaremba et al. [26] explored the relationship between government policy responses to the COVID-19 pandemic and stock market volatility. They gathered data about seven non-pharmaceutical interventions from 67 countries and concluded that information campaigns and the cancellation of public events were the major accelerators of market volatility.

Kizys et al. [18] used 72 stock market indices from both developed and emerging economies and tested whether government policy responses to the COVID-19 pandemic could mitigate investors' herding behavior. Overall, their results point to the herding phenomenon in international capital markets, but policy responses reduced such behavior. Ozili and Arun [27] tested the impact of government measures during the COVID-19 epidemic on the performance of leading market indices on four continents: UK, US, Japan and South Africa. They revealed that the increasing number of lockdown days, monetary policy decisions and international travel restrictions severely affected the level of general economic activity and the closing, opening, lowest and highest stock prices of major stock market indices. In contrast, the restrictions imposed on internal movement and increased fiscal policy spending had a positive impact on the level of economic activity.

More recently, Huang et al. [28] tested the effects of COVID-19 government policies on the hospitality labor market in the U.S. They found that closure policies were associated with a 20% - 30% reduction in non-salaried workers in the hospitality industry. Furthermore, the number of daily confirmed cases adversely affected the hospitality sector's labor market.

Previous studies such as the work of Adda [16] explored the unintended consequences of economic activity on the spread of infections and assessed the efficiency of measures that limit interpersonal contacts in France. They found that policies reducing interpersonal contacts such as closing schools and public transportation significantly reduced the spread of disease, although they were not cost-effective. Pennathur et al. [20] explored the impact of the U.S. government interventions in response to the subprime financial crisis on the stockholder returns of banks, savings and loans firms, insurance companies, and REITs. They found that interventions reduced the wealth and increased the risks of financial institutions. Ding et al. [11] used

an event study approach to examine the Chinese stock market's response to the lockdown restrictions imposed on Hubei province in light of COVID-19. In general, this response was negative. Furthermore, firms that had a great deal of exposure to Hubei earned significantly lower returns following Hubei's lockdown measures.

Chen et al. [17] examined the effect of the SARS epidemic on the stock prices of seven Taiwanese hotel stocks using an event-study approach. They reported that these firms suffered from steep declines in their earnings and stock prices during the SARS outbreak. Ru et al. [14] explored the cumulative abnormal returns during the COVID-19 epidemic for two groups of stock markets: countries that had experience with SARS, and countries that did not. They documented a stronger negative response in the markets in the experienced countries.

Using daily data about confirmed cases and deaths from the coronavirus and stock market returns from 64 countries, Ashraf [10] found that stock markets responded negatively to the growth in confirmed COVID-19 cases. Furthermore, there was a weak relationship with the number of deaths. Al-Awadhi et al. [9] explored companies included in the Hang Seng Index and Shanghai Stock Exchange Composite Index during the COVID-19 pandemic, and established a significant negative relationship between both the daily growth in total confirmed cases and the daily growth in total deaths caused by COVID-19. Recently, Goodell et al. [13] investigated the abnormal returns of 49 U.S. industry portfolios around COVID-19 news announcements. They documented that on February 26, 2020, when the first domestic case was confirmed in California, 15 industries reacted negatively to this news. The industries most sensitive to the news around February 26, 2020 were utilities, services and restaurants, hotels, and motels. Gerding et al. [12] examined stock market reactions to the COVID-19 outbreak around the world. They found that the market response was more aggressive in countries with a higher debt to GDP ratio. Finally, Ding et al. [29] evaluated the degree to which pre-crisis corporate conditions affected stock price behavior with respect to the COVID-19 epidemic. They reported that stock fluctuations were more moderate in firms that engaged in more CSR activities, and had more cash, less debt, and larger profits. They also indicated that stock prices were less exposed to the negative of COVID-19 if they had global supply chains and customer locations, and less entrenched executives. To summarize, we will add to the standing literature by investigating the impact of various U.S. government intervention measures on the tourism industry.

## COVID-19's impact on the hospitality industry

The COVID-19 pandemic had an unprecedented negative impact on the hospitality industry. According to a report published by the American Hotel and Lodging Association (https://www.ahla.com/sites/default/files/recessiondepression_0.pdf), the expected US hotels losses are nearly $83.7 ($51.2) billion in room revenue in 2020 (2021), compared with 2019, while job losses in 2020 (2021) are projected to be nearly 630,000 (546,000). In addition, about half of hotel markets, representing 72% of hotel rooms in the US, are still in a recession or depression. These numbers indicate that most of the hotel industry has a long road to recovery, especially when considering that an occupancy rate of 35% or lower makes it impossible for many hotels to stay open. In Fact, individual hotels and major operators are projecting occupancies below 20% (https://hoteltechreport.com/news/tourism-industry-statistics#hotels).

Similarly, the general state of the travel and tourism industry is also under a great threat. According to the Economic Impact Report by the World Travel and Tourism Council [30], prior to the pandemic, the travel and tourism sectors, both directly and indirectly, accounted for 1 in 4 of all new jobs created around the world, 10.6% of all jobs (334 million), and 10.4% of global GDP (US$9.2 trillion). In 2020, 62 million jobs were lost, representing a drop of

18.5% (62/334). The threat of job losses is continuing as many jobs are currently supported by government retention schemes and reduced hours, which without a full recovery in this sector, could be lost.

A careful mapping of the literature shows that several papers have reported supporting evidence for COVID-19's adverse effect on the performance of the hospitality sector. Hao et al. [31] reviewed the overall impact of the pandemic on China's hospitality industry—the country where the health crisis began. The industry witnessed a sharp decline in hotel occupancy rates and a loss of over US$9 billion in revenue. About 74% of the hotels in China were closed in January and February 2020 for an average period of 27 days. Furthermore, from January 14 to 28, the occupancy of the hotels dropped from around 70% to 8% and remained under 10% in the following 28 days. As a result, the hotel and tourism industry reduced their number of employees, leading to a significant drop in cash flow and revenue. Zheng et al. [32] studied the phenomenon of "travel fear" in China. They reported that perceptions about the severity of the threat and the susceptibility to it can cause "travel fear," which leads to protective behaviors with regard to travel decisions. Furthermore, "travel fear" can evoke different strategies that increase people's psychological resilience and adoption of cautious travel behaviors. Villace-Molinero et al. [33] explored perceptions about travel risks during the pandemic and proposed measures to improve traveler confidence based on the issue-attention cycle. Based on a survey conducted in 46 countries and a qualitative study in which 28 international hospitality experts were interviewed, the authors concluded that in a pandemic scenario, confidence in communications from the local government about personal safety and security are the main factors people consider when making travel decisions.

Lee et al. [34] tested the impact of COVID-19 on hospitality stock returns in China. They argue that the increasing uncertainty about the COVID-19 outbreak has made the Chinese stock market more turbulent and less predictable. Using a structural vector auto regression (SVAR) framework, they examined the link between the COVID-19 outbreak, macroeconomic fluctuations and hospitality stock returns in China. Their results hint that macroeconomic fluctuations and hospitality stock returns are significantly affected by shocks from the COVID-19 outbreak. Crespí-Cladera et al. [35] used a stress methodology to estimate the potential performance vulnerability for Spanish hospitality firms. They demonstrated that almost 25% of their sample firms are exposed to financial distress when operational income decreases 60%. They also found that the majority of such firms are generally small ones, which would also suffer from solvency problems. When hospitality firms' revenues drop 80%, the predictions show that 32% of firms would be in financial distress. Rodríguez-Antón and Alonso-Almeida [36] reported that the performance of the hospitality industry in Spain has been severely damaged as a result of COVID-19. More specifically, they noted that in the first seven months in 2020, the total hotel overnight stays in Spain declined from 184.7 million in 2019 to nearly 46.4 million. In addition, the pandemic reduced the number of new hotel openings (−22.02%) and the number of employees hired (−30.94%) in March 2020. Finally, a recent study of Clark et al. [37] documented the negative impact of COVID-19 on the stock performance of hospitality firms. They estimated negative mean cumulative abnormal returns of −17.54% for 54 publicly traded hospitality firms from 23 different countries. Restricting their sample to the US or Japan yielded negative cumulative abnormal returns of −29.67% and −10.68%, respectively.

Other studies have examined the impact of previous pandemics, such as the severe acute respiratory syndrome (SARS), on the performance of the hospitality industry. Chien and Law [38] showed that the outbreak of SARS in March 2003 had a strong negative impact on the hotel business in Hong Kong. The occupancy rates of many hotels in Hong Kong fell to 10% or less in March and April 2003, which normally is the peak season. Similarly, Hendersom and

Ng [39] reported statistics from the Singapore Tourism Board confirming the severity of the impact of SARS on the hospitality sector there. According to the report, the average hotel occupancy rate for the second quarter of 2003 was 21%, compared with 74.5% for the previous year, and average room rates dropped by 18.8%. In addition to this report, they also surveyed hotels in Singapore to estimate the economic loss resulting from SARS. They noted that in their own surveyed hotels the average occupancy rates were also relatively low, in the range of 27.7% to 42.3%.

Kim et al. [40] tested the effect of SARS on the Korean hotel industry. They examined six Korean hotels and reported that the occupancy rate dropped nearly 14% from February to July 2003. They argued that the reason seemed to be that inbound tourists saw Korea as an unsafe tourism destination within the territory of the SARS-affected Asian Pacific zone. Revenue per available room (RevPAR) during the three months from April to June was 215,849 won (US $180) in 2002, whereas it was 115,676 won (US$96) in 2003, a 100,173 won (US$83) difference. Finally, there was a 16% drop in profit margins from April through June in 2003 compared to the same period in 2002. The average rate per room also declined by 19% as hotels attempted to cope with the sharp decline in demand. Tew et al. [41] used a questionnaire designed to investigate, among other points, the impact of SARS on hotel performance in Canada. Respondents were asked to assess the impact that the SARS crisis had on their hotel's performance. Over 82% of respondents reported that SARS had an extremely negative or very negative impact on their hotel's performance. In addition, Tew et al. (2008) [41] also reported that the Niagara Falls region experienced a loss of over 122,000 room nights in the second quarter of 2003. This loss translated into a loss of $19 million in room revenue. Finally, Chen et al. [17] examined the effect of the SARS epidemic on Taiwanese hotel stock price movements using an event-study approach. They showed that for seven publicly traded hotel companies there were steep declines in earnings and stock prices during the SARS outbreak period. More specifically, they reported that in April 2003 hotel companies experienced drops in earnings in the range of −49.81% to −11.14%. They also showed that these drops worsened significantly when extending the period examined for two months (April–June, 2003), with reductions varying from −76.89% to −20.00%. Finally, they calculated the cumulative abnormal returns of stocks in this sector during 10 and 20-day windows from the day of the SARS outbreak. The negative returns they found were also robust using different types of models to estimate the abnormal returns.

Importantly, COVID-19 has not only directly affected the hospitality industry performance, but also created collateral damage that might indirectly harm it. The literature suggests several possible additional factors behind the poor performance of the hospitality industry that might delay its future recovery. These effects are evident in the labor force (Jung et al. [42]), its mental health (Yan et al. [43]), and the willingness to travel and the spreading of fake news (Alvarez-Risco et al. [44]).

To summarize, these studies show that in addition to the negative effect that government interventions usually have on financial markets, COVID-19 also had various detrimental effects on the hospitality industry. Therefore, combining these two pieces of evidence, we might expect that the impact of government interventions on the hospitality industry would also be negative.

## Data and methodology

### Sample construction and data sources

Our sample consists of the daily log returns of stock portfolios consisting of firms operating in the hospitality industry in the following COMPUSTAT SIC codes: Retail–eating places (5800–5819), Restaurants, hotels, motels (5820–5829), Eating and drinking places (5890–5899),

Hotels, other lodging places (7000–7000), Hotels and motels (7010–7019), Membership hotels and lodging (7040–7049) and Services–linen (7213–7213). We refer to these related industries collectively as the hospitality industry.

We use market prices as a proxy for the overall state of the hospitality industry as well as for the other related sectors. This approach might have limitations, albeit temporary ones, which stem from behavioral biases. Nevertheless, using market prices is still a prevalent method that reflects the present value and state of securities. Additionally, we retrieved data for the log returns of stocks in nine other related industries (Food Products, Candy & Soda, Beer & Liquor, Entertainment, Consumer Goods, Apparel, Personal Services, Transportation and Retail). All firms in each portfolio are traded on the NYSE, AMEX, and NASDAQ exchanges. The data are publicly available on Kenneth French's website and cover the period of December 31, 2018 to April 30, 2020. They include 336 daily returns for each industry portfolio and a total sample of 3,360 daily observations (http://mba.tuck.dartmouth.edu/pages/faculty/ken.french/data_library.html). We also retrieved data from Kenneth R. French's data library about the performance of a market portfolio (MARKET). According to French's definition, the market portfolio consists of the value-weighted returns of all CRSP firms incorporated in the US and listed on the NYSE, AMEX, or NASDAQ (See also [45] for a complete description).

Table 1 as well as Fig 1 present information about the cumulative returns of the hospitality and other related industries for January 2020 to April 2020, compared to the general market's performance. As can be seen, the overall negative performance is not limited to the hospitality industry. In fact, several other industries are associated with excess negative returns compared with the hospitality industry. Note too that the worst month for all of these industries was March 2020, when most of the interventions occurred. During this month negative returns abounded. Indeed, the hospitality industry lost about 45% of its cumulative market value. The recovery in the market value of the various sectors, including the hospitality industry, took place in April 2020. However, nearly all industries ended the period with a substantial decrease in their market value.

Our empirical discussion also utilizes Baker et al.'s measure of uncertainty due to infectious diseases [46]. The data come from their Economic Policy Uncertainty website and are available since 1985 (http://policyuncertainty.com/infectious_EMV.html). In designing this index,

**Table 1. The performance of U.S. hospitality stocks and those in closely related industries during COVID-19.**

| # | Cumulative Return | JAN. | FEB. | MAR. | APR. |
|---|---|---|---|---|---|
| 1 | MARKET | 0.09 | -8.02 | -18.58 | -5.10 |
| 2 | Food Products | -4.58 | -13.23 | -12.49 | -4.71 |
| 3 | Candy & Soda | 5.48 | -1.45 | -16.55 | -0.46 |
| 4 | Beer & Liquor | -0.72 | -8.89 | -25.90 | -4.50 |
| 5 | Entertainment | 0.53 | -10.05 | -43.39 | -14.18 |
| 6 | Consumer Goods | -2.90 | -14.24 | -36.03 | -17.43 |
| 7 | Apparel | -7.37 | -19.98 | -49.94 | -29.71 |
| 8 | Personal Services | -5.43 | -9.97 | -34.74 | -15.01 |
| 9 | Transportation | -4.96 | -16.42 | -35.83 | -22.01 |
| 10 | Retail | -7.12 | -16.48 | -40.39 | -12.63 |
| 11 | HOSPITALITY | 0.34 | -8.47 | -44.59 | -11.37 |

Notes: The table reports the cumulative returns of a portfolio consisting of U.S. hospitality firms such as restaurants, hotels and motels (#11) during the four months since the outbreak of COVID-19 in January 2020. We also report the returns of stocks in additional industries (#2 through #10) associated with the hospitality industry. The first row (#1) presents the performance of the overall market, represented by all NYSE, AMEX, and NASDAQ firms. For the sake of brevity, complete descriptive statistics are not provided but are available upon request.

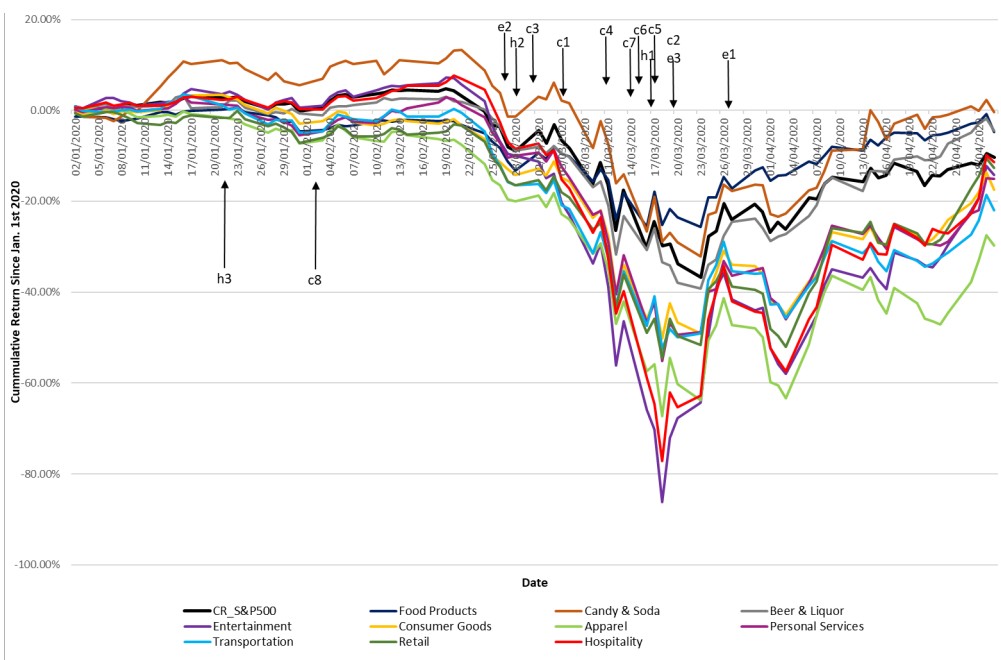

**Fig 1. Comparison of the cumulative returns of the S&P 500, the hospitality industry and its related industries indexes A**. The figure depicts the cumulative returns of the S&P 500, the hospitality industry and the indices of numerous related industries since the outbreak of COVID-19 in January 2020. "c," "e" and "h" denote closures, economic and health measures, respectively.

Baker and colleagues created an index based on the frequency with which various terms appeared daily in approximately 3,000 US newspapers. They classified these articles into three categories: E: economic, economy, financial; M: stock market, equity, equities, Standard and Poors; and V: volatility, volatile, uncertain, uncertainty, risk, and risky. Finally, these articles must also mention one or more terms related to epidemic, pandemic, virus, flu, disease, coronavirus, MERS, SARS, Ebola, H5N1, or H1N1. The resulting counts were scaled by the count of all articles on the same day. Fig 2A and 2B plot the evolution of epidemic-based uncertainty. As can be seen, uncertainty skyrocketed upwards multiple times above the average. The maximum value reached was 112.93, recorded on March 15, 2020.

Fig 2A provides a combined snapshot of the uncertainty due to infectious diseases and the performance of the hospitality portfolio, NASDAQ and S&P500. The overall picture illustrates a clear inverse relationship. A high level of uncertainty is accompanied by a decrease in the hospitality returns (as well those of the market portfolios), and vice versa.

## Event study

Event study methodology explores the response to a specific event by assessing whether it creates abnormal stock returns that can be attributed to new information released. Therefore, in using this method, our first step was determining the event of interest and defining the length of the event's window. To do so, we collected the dates of the government responses to COVID-19 from Hale et al.'s database [1] (See the Oxford COVID-19 Government Response Tracker on https://covidtracker.bsg.ox.ac.uk/). This database is constructed from publicly available sources such as news articles and government press releases and briefings. We also identified March 19, 2020 as another significant date because it marked the day when President Trump signed a $100 billion economic aid package.

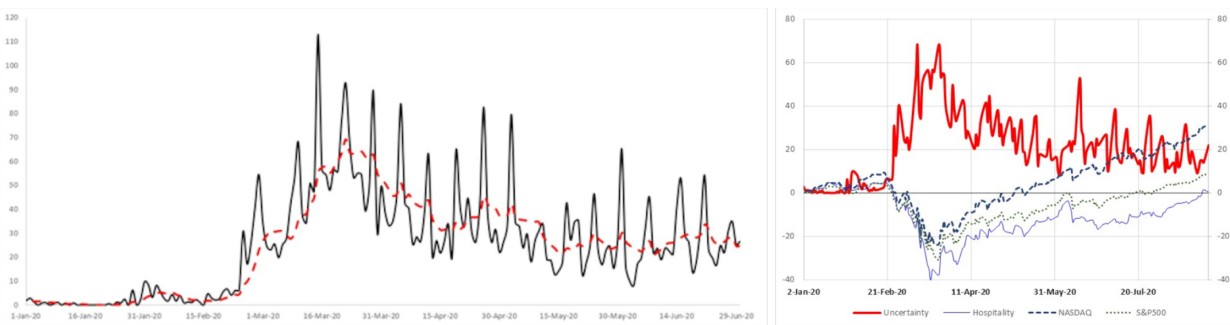

**Fig 2. Evolution of the pandemic-driven uncertainty and market indices.** (A) The dotted line is a one-week moving average of the index. The figure indicates that starting from the last week of February 2020, pandemic-driven uncertainty skyrocketed to record values. The figure includes 7-day-a-week observations. (B) The solid bold red line is Baker et al.'s daily index of uncertainty due to infectious diseases [46]. This uncertainty measure considers the frequency of U.S. newspaper articles that include terms related to economy, risk, financial market, uncertainty and epidemic. The uncertainty index observations correspond with the trading days on the stock exchange. The correlation between this form of uncertainty and variations in returns is 0.46 (t-stat = 4.66). Overall, the message of the figure is that high levels of uncertainty do not bode well for a quick recovery in the hospitality sector.

Table 2 presents the list of U.S. government interventions with a definition for each intervention variable. The responses appear chronologically to illustrate how the government's actions evolved over the full period of COVID-19's spread.

Next, we defined the length of the event window in which we examined the behavior of the equity prices of stocks in the hospitality sector and those in related industries. In fact, there is no golden number for the length of the event window. On one hand, a longer window relying on a large number of daily returns may reveal more information. On the other hand, other events occurring during a long-time window might contaminate the results. Given the proximity of events, we focused on a relatively narrow timeframe to minimize the possibility of confounding events. We adopted this approach to isolate the impact of each intervention, and because the COVID-19 period was followed by many events apart from the interventions. In addition, using a long timeframe might increase the chances of contemporaneous and intertemporal residual correlations that could result in underestimates of the standard errors [47]. Brown and Warner [48, 49], as well as McWilliams and Siegel [50] argued that a long event window also reduces the power of the test statistics and leads to false interpretations. Therefore, we chose an event study that spans three days, that is, from $t_{-1}$ to $t_{+1}$ days, which should cover the impact of government interventions on equity indices.

The center of each event study ($t_0$) is the announcement day (henceforth, day zero). Using Hale et al.'s [1] database, we defined day zero as the first day on which the intervention was announced to the public. If the announcement was made on a day when the exchange was closed, we defined day zero to be the first day when the exchange was open again. For instance, if the intervention was publicly announced on Saturday or Sunday, day zero would be the following Monday.

The subsections provide a detailed explanation of how we measured the returns, expected returns and abnormal returns as well as additional information concerning government interventions.

## Abnormal returns (AR)

To estimate abnormal returns, we followed several studies that use the event study methodology in the hospitality and other related industries (e.g., [3, 17, 51, 52]). We extracted the residual returns from the well-known Capital Asset Pricing Model (CAPM, by Sharpe, [53]), which

**Table 2. Description of U.S. interventions.**

| Event Date | ID | Intervention | Intervention Definition |
|---|---|---|---|
| 21/01/2020 | h3 | Contact tracing | Record if the U.S. government is doing contact tracing |
| 02/02/2020 | c8 | International travel | Record restrictions on international travel |
| 27/02/2020 | e2 | Debt contract relief | Record if the U.S. government is freezing financial obligations (e.g. stopping loan repayments, preventing services such as water from being cut off, or banning evictions) |
| 28/02/2020 | h2 | Testing policy | Record the presence of COVID-19 test policy |
| 01/03/2020 | c3 | Cancel public events | Record cancelling public events |
| 05/03/2020 | c1 | School closing | Record closings of schools and universities |
| 11/03/2020 | c4 | Restrictions on gatherings | Record the cut-off size for bans on private gatherings |
| 14/03/2020 | c7 | Domestic travel | Record restrictions on internal movement |
| 15/03/2020 | c6 | Stay at home requirements | Record orders to "shelter-in-place" and stay home. |
| 16/03/2020 | h1 | Public info campaign | Record the presence of public information campaigns |
| 17/03/2020 | c5 | Close public transport | Record closing of public transportation |
| 19/03/2020 | c2 | Work place closing | Record closing of workplaces |
| 19/03/2020 | e3 | Aid Package | President signs a $100 billion aid package |
| 27/03/2020 | e1 | Income support | Record if the U.S. government is covering the salaries of or providing direct cash payments, universal basic income, or similar, to people who lose their jobs or cannot work. (Includes payments to firms if explicitly linked to payroll/ salaries) |

Notes: The table reports the intervention decisions made by the U.S. government during the COVID-19 epidemic. The main source is Hale et al. [1], which tracks government responses in countries around the globe to the COVID-19 epidemic. The interventions are sorted by their chronological appearance during the epidemic. The first column presents the ID of each intervention, where e, c and h are abbreviations for economic, closure and health government measures, respectively. We also defined an additional event that corresponds to e3, when President Trump signed a $100 billion aid package as part of the war against COVID-19.

appears in Eq (1):

$$\varepsilon_{CAPM,t} = R_{p,t} - (\hat{\alpha}_p + \hat{\beta}_p R_{mt}). \tag{1}$$

$R_{p,t}$ in Eq (1) is the rate of return of stocks in industry portfolio $i$ on day $t$, and $R_{mt}$ is the market portfolio return, including all NYSE, AMEX, and NASDAQ firms on day $t$. The parameters $\hat{\alpha}_p$ and $\hat{\beta}_p$ are the constant and systematic risk estimated parameters of stocks in industry portfolio $i$, respectively, and $\varepsilon_{CAPM,t}$ is the error term.

The term in parentheses on the right-hand side of Eq (1) is the expected normal return. The error term is the industry-specific component or the unexpected return, which can be attributed to the new information released such as the intervention. We explored each point in time separately from the market performance.

We estimated the coefficients' parameters in Eq (1) using a pre-event estimation period (December 31, 2018-December 30, 2019) consisting of 252 trading days before the outbreak of COVID-19, as on December 31, 2019, the WHO was informed of cases of pneumonia of an unknown cause in Wuhan City, China. Finally, we aggregated the abnormal returns (CAR) and computed the t-statistics, following Brown and Warner [49].

## Causality tests

The second step of our methodology attempts to link uncertainty as a potential driver of the variations in the returns of the hospitality stocks. Therefore, we also utilized the Granger causality test to examine the ability of the uncertainty due to infectious diseases to explain the variations in the returns of the hospitality sector in the COVID-19 period. Technically speaking, this procedure leads to a statistical hypothesis test for determining whether a given time series is useful for forecasting another one. Hence, this procedure, by and large, is closely related to the terms "cause" and "effect." We used the Granger causality procedure to examine whether time series x "Granger-causes" time series y [54]. If this relationship holds true, then lagged values of x should contain information that helps predict y. This relationship reads as follows.

$$\begin{pmatrix} y_t \\ x_t \end{pmatrix} = \begin{pmatrix} \beta_{10} \\ \beta_{20} \end{pmatrix} + \begin{pmatrix} \beta_{11} & \alpha_{11} \\ \alpha_{21} & \beta_{21} \end{pmatrix} \begin{pmatrix} y_{t-1} \\ x_{t-1} \end{pmatrix} + \begin{pmatrix} \beta_{12} & \alpha_{12} \\ \alpha_{22} & \beta_{22} \end{pmatrix} \begin{pmatrix} y_{t-2} \\ x_{t-2} \end{pmatrix} + \cdots + \begin{pmatrix} \beta_{1k} & \alpha_{1k} \\ \alpha_{2k} & \beta_{2k} \end{pmatrix} \begin{pmatrix} y_{t-k} \\ x_{t-k} \end{pmatrix} + \begin{pmatrix} u_{1t} \\ u_{2t} \end{pmatrix}, \quad (2)$$

where k denotes the length of the lag, and $u_i$ (i = 1,2) is the disturbance term. This model allows us to test whether lagged values of one variable, say X, helps explain current values of another one, say Y. Formally, we can infer such an outcome if the hypothesis that $\alpha_{11} = \alpha_{12} = \ldots = \alpha_{1k} = 0$ is rejected. In other words, if we do not reject this hypothesis, then variable X does not Granger-cause Y.

## OLS and regression analysis

Lastly, after testing the potential relationship between hospitality stock returns and uncertainty, we explore the uncertainty levels around the interventions themselves. To do so, we employ OLS estimations, which link government interventions with uncertainty. Such examination will allow us to reveal whether government interventions induce uncertainty, fear and anxiety, which will be then translated to hospitality stock returns performance. To obtain heteroscedasticity and autocorrelation consistent covariance matrix estimates, we employed Newey and West's [55, 56] estimation method.

## Empirical findings

Table 3 presents a summary of the effect of each measure on the hospitality industry and related industries sorted by the type of intervention. We present and discuss the findings chronologically to understand how the government's actions evolved throughout the spread of COVID-19.

In parallel, Fig 1 plots a comparison of the cumulative returns of the market portfolio (S&P 500), the hospitality equity portfolio, and other related industries relative to various event dates associated with COVID-19's outbreak. The figure illustrates the high degree of co-movement between the hospitality and other related industries, and the shocks to these industries relative to the market portfolio.

As the table indicates, most of the events are associated with a significantly negative response to U.S. government interventions. The only exception is the aid package (e3), announced on March 19, 2020, which mostly had a significant positive effect.

As illustrated in Fig 1, between January 2, 2020 and February 20, 2020, the cumulative returns of the hospitality industry portfolio (depicted in red) rose moderately, implying that the government's earlier actions had no influence on the performance of hospitality companies. More specifically, the interventions of *contact tracing* (h3) and restricting *international travel* (c8) did not have any statistically significant effect on the hospitality industry's performance and most of the related industries. One explanation for this result might be the

**Table 3. The impact of government interventions on the hospitality industry and industries closely related to it.**

| Panel A | CAR[-1,+1] | | | |
|---|---|---|---|---|
| | **21/01/2020** | **02/02/2020** | **27/02/2020** | **28/02/2020** |
| | **h3** | **c8** | **e2** | **h2** |
| | **Contact tracing** | **International travel** | **Debt/contract relief** | **Testing policy** |
| **Food Products** | 0.552 | **-2.004**[**] | **-3.396**[***] | -0.807 |
| **Candy & Soda** | -0.426 | 3.304 | -2.762 | -0.204 |
| **Beer & Liquor** | -1.103 | -0.153 | -3.005 | -3.948 |
| **Entertainment** | 0.373 | -0.213 | **-2.149**[*] | **-3.810**[***] |
| **Consumer Goods** | -0.205 | -1.066 | 0.304 | -0.883 |
| **Apparel** | -1.637 | -0.835 | 1.604 | -1.378 |
| **Personal Services** | -1.648 | -0.397 | -0.206 | -1.460 |
| **Transportation** | **-3.296**[**] | -1.085 | -0.257 | **-3.092**[**] |
| **Retail** | -0.038 | -1.163 | -0.022 | -2.106 |
| **HOSPITALITY** | -0.330 | -0.433 | **-5.360**[***] | **-4.195**[***] |
| Panel B | CAR[-1,+1] | | | |
| | **01/03/2020** | **05/03/2020** | **11/03/2020** | **16/03/2020** |
| | **c3** | **c1** | **c4** | **c7, c6, h1** |
| | **Cancel public events** | **School closing** | **Restrictions on gatherings** | **Mixed (1)** |
| **Food Products** | 0.048 | 1.182 | **-3.085**[***] | **4.581**[***] |
| **Candy & Soda** | 3.383 | -0.056 | -0.370 | **-4.960**[*] |
| **Beer & Liquor** | -3.205 | -0.063 | **-7.447**[***] | 3.665 |
| **Entertainment** | **-5.908**[***] | **-7.105**[***] | **-14.561**[***] | **-16.29**[***] |
| **Consumer Goods** | **-2.123**[**] | -0.331 | **-8.096**[***] | **-4.089**[***] |
| **Apparel** | **-2.523**[*] | -1.313 | **-4.263**[***] | **-11.94**[***] |
| **Personal Services** | -1.388 | **-2.528**[**] | **-8.879**[***] | **-4.002**[***] |
| **Transportation** | **-3.045**[**] | -1.902 | 0.123 | -0.872 |
| **Retail** | -2.718* | -0.124 | **-6.648**[***] | **-5.127**[***] |
| **HOSPITALITY** | **-3.394**[***] | **-6.404**[***] | **-10.490**[***] | **-21.98**[***] |
| Panel C | CAR[-1,+1] | | | |
| | **17/03/2020** | **19/03/2020** | **27/03/2020** | |
| | **c5** | **c2, e3** | **e1** | |
| | **Close public transport** | **Mixed (2)** | **Income support** | |
| **Food Products** | -0.855 | -1.228 | **2.872**[***] | |
| **Candy & Soda** | **-6.186**[**] | -3.812 | 1.859 | |
| **Beer & Liquor** | -1.404 | **-5.656**[**] | **4.716**[*] | |
| **Entertainment** | **-30.59**[***] | **9.292**[***] | **-9.379**[***] | |
| **Consumer Goods** | **-6.603**[***] | 1.772 | **-3.217**[***] | |
| **Apparel** | **-12.09**[***] | **5.147**[***] | **-6.873**[***] | |
| **Personal Services** | **-12.97**[***] | 0.066 | -0.725 | |
| **Transportation** | **-3.341**[**] | 0.822 | **-8.262**[***] | |
| **Retail** | **-5.862**[***] | **5.432**[***] | **-7.772**[***] | |
| **HOSPITALITY** | **-28.72**[***] | **5.593**[***] | **-8.252**[***] | |

Notes: The table reports the industry's response to government measures according to each type of intervention. CAR measure the response of each industry by aggregating the abnormal returns from the well-known market model by Sharpe [53], in a time window that spans $t_{-1}$ to $t_{+1}$. Following Brown and Warner [49] t-statistics are computed for AR and CAR. Mixed (1) [Domestic travel, Stay at home requirements and Public info. Campaign] and Mixed (2) [Workplaces closing and Aid Package] are special events in which two or more interventions shared the same zero day. Significant CAR appear in bold, and "***," "**" and "*" denote statistical significance at the 1%, 5% and 10% levels, respectively.

common belief in the early stages of the COVID-19 pandemic that it would not spread to the U.S. Naturally, people were still optimistic that COVID-19 was just another form of the flu that would have a minor effect on the economy. Apart from the explanation for this relatively passive market response coming from the fact that this date was part of the early stage of the outbreak, many may have believed the claim of China's Foreign Ministry on the very same day that the U.S. government was reacting inappropriately to the outbreak and spreading fear by enforcing travel restrictions.

From this point in time, which we can call the realization period, the U.S. engaged in more frequent and extensive interventions, underscoring the severity of COVID-19. In fact, the period from February 21 to March 18 is characterized by a sharp decline in the hospitality industry's performance as well as on other related industries. It was during this period that a series of interventions were imposed that also had an increasingly negative impact on the hospitality industry and the economy as a whole. During this period of time, even the economic intervention of debt/contract relief (e2), which postponed debt payments, had a negative effect (CAR = –5.36%) on the hospitality industry and also for the food (–3.396%) and entertainment (–2.149%) industries. These findings raise the possibility that the market was overwhelmed by negative sentiment. However, it may also reflect that the expectations about economic aid and substantial government support were much more meaningful. As evident in its cumulative negative returns, the hospitality industry regarded the announcements about debt/contract relief (e2) as insufficient, especially, due to the opinion that the hospitality sector would be the first to be hurt and the last to recover. In addition, according to the Oxford database, the debt/contract relief intervention in the U.S. provided only limited relief because it just postponed debt payments with no clear indication about real economic recovery, or any guidance about plans for the long term. When we delved deeper, we also reveal that February 26, 2020 was a significant milestone. It was the date on which the first U.S. coronavirus case of unknown origin was documented, which possibly induced negative response.

The next intervention of the U.S. government occurred in the health field and involved the announcement about a COVID-19 testing policy. The policy sets the criteria for who is to be tested for COVID-19. Intuitively, such a measure should be interpreted as good news. However, the overall response of the hospitality industry to the U.S. government's *testing policy* intervention (h2) was negative (CAR = –4.195%). As for the related industries, the findings show mainly insignificant negative responses on the event date. The only exceptions are entertainment (–3.810%) and transportation (–3.092%). One explanation for this counterintuitive outcome might be that testing was offered only to those who had apparent symptoms and met specific conditions. Therefore, the industry regarded this as a minor and insufficient step in stopping the pandemic.

Several days afterwards, on Monday, March 1, 2020, the *canceling of public events* (c3) was imposed. The results indicate a clear negative effect on the hospitality industry's performance; the cumulative abnormal returns (–3.394%) were negative and statistically significant. Actually, in most of the industries, there were statistically significant and consistently negative abnormal returns. These findings highlight the powerful effect of cancelling public events on this industry, which may be a main source of the hospitality companies' income.

This major negative response may also be a result of two additional events that occurred on the same weekend when the stock market was closed. The first event was the official record of the first coronavirus death in the U.S., and the second was an extension of the travel restrictions to include Italy and South Korea. These events probably intensified the negative market response. Apparently, now, the market starts to comprehend that the coronavirus is here to stay, and is far from being a trivial episode. In addition, canceling public events is a real

business obstacle, and with the overall comprehension of the new challenging and unclear business activity, it might explain the overall negative response.

A quick glance at Fig 2A and 2B supports this notion. The figures show that the measure of COVID-induced economic uncertainty in Baker et al.'s index [46] started to deviate from the mean values observed prior this period implying that the media and newspapers devoted extensive attention to the pandemic and to its economic consequence. In this spirit, prior works have documented that media-driven pessimism–fueled by the outbreak of a pandemic– has a remarkable impact on stocks exposed to intense media coverage (e.g., [57]).

From this point forward, the U.S. engaged in more frequent and severe interventions. The findings show the clear negative response of both the hospitality sector and most of its closely related industries to these announcements. On March 5 and March 11, 2020, the closure interventions were extended and included new restrictions such as *school closings* ($c_1$) and *restrictions on gatherings* ($c_4$). The results in Table 3 demonstrate an evident adverse impact that was much more aggressive in terms of its magnitude. Specifically, for the hospitality industry, the abnormal returns in the wake of school closings were –3.786%, while restrictions on gatherings had a total negative effect of –5.550%. Note that on the evening of March 11, 2020, President Trump expanded the travel restrictions on foreign travelers, banning entry for the next 30 days from 26 countries in Europe except for Ireland and the United Kingdom. This announcement probably intensified the negative market reaction on March 12, 2020. The negative market reaction was also reflected in the significant cumulative abnormal returns for the hospitality industry (–10.490%), and for the entertainment and personal services industries (–14.561% and –8.879%, respectively).

The next intervention (mixed (1)) was an even more significant one, given that it involved several measures of different types. The mixed (1) intervention involved three events: domestic-travel restrictions ($c_7$) on local movement between cities and regions, stay at home requirements ($c_6$) and public information campaigns ($h_1$) to raise awareness of the coronavirus. These steps were publicly announced between Saturday, March 14, 2020 and Monday, March 16, 2020. As the reported results in Table 3 illustrate, the combined effect of these interventions had a particularly strong impact on the hospitality industry's performance (CAR = –21.98%), indicating the severe effects of the stay at home and public transportation restrictions which also spilled over to the entertainment (CAR = –16.29%) and apparel (CAR = –11.94%) industries.

On March 17, 2020, the closing of public transportation ($c_5$) was the most significant event affecting the stock prices of the hospitality industry. The market value of the hospitality industry plunged almost 29% in three days (CAR = –28.72%), while the entertainment industry lost almost one-third of its market value in just three days (CAR = –30.59%). The record level of uncertainty due to the pandemic observed on this day appears in Fig 2A and reflects the tense atmosphere in the economy.

There are two possible explanations for this clearly significant tendency for negative returns following the government's interventions, rooted in two different, but somewhat related, reasons. The first is the ambiguity about the government's future fiscal steps that increased the uncertainty in the stock market [23]. The public finance literature has established that frequent changes in government policy may have direct implications for the stability of macroeconomic variables such as GDP, consumption and debt (e.g., [25]). The second reason is related to the uncertainty surrounding the pandemic itself and the unclear picture in its initial stages. This uncertainty includes the ambiguity about the real consequences for the economy, the time required for economic recovery, the rapidity of the spread of the infection and its lethality. All of these factors offset the impact of the interventions, or at least postponed their immediate effect until a bit later.

Another set of several interventions imposed together are described in the mixed (2) interventions in Table 3. The mixed (2) intervention involved two events. On March 19, the government announced a new restriction that required all but essential workplaces to close and people to work from home if they could (c2). In addition, on Wednesday evening, March 18, 2020 when the exchange was closed, President Trump signed a $100 billion coronavirus emergency aid package, which included provisions for emergency paid leave for workers as well as free testing (e3). Table 3 shows that the good news about the assistance package announced on March 18 during the evening overcame the negative news of workplace closures. This result is both interesting and important given that according to the American Hotel and Lodging Association, job losses in the industry are expected to be extensive. In 2020, job loss was projected to be nearly 630,000, while in 2021 job loss were still expected to be around 546,000. Given that about half of hotel markets, representing 72% of hotel rooms in the US, are still in a recession or depression, it seems that providing economic aid might help calm the adverse effect of the expected job losses and the negative impact of imposed closures. Finally, the economic assistance had a major positive impact not only on the hospitality industry but also on several other industries such as entertainment, apparel and retail (CAR = +9.292%, +5.147% and 5.432%, respectively).

Indeed, the announcement about closing workplaces probably was expected, given all of the former restrictions, especially after the stay at home requirement made only three days before. Note that the aid package (e3) was a turning point in the behavior of the hospitality industry performance. It ended the one-month period of decline and started a positive upward trend that continued, though with fluctuations, through April 30, 2020, when the cumulative returns of the hospitality industry portfolio matched the cumulative returns of the S&P 500 index (see Fig 1).

Lastly, and in line with the Keynesian theory that countercyclical fiscal policy actions such as lower taxes or more fiscal spending under adverse economic conditions may help the economy recover (e.g., [58]), on Friday, March 27, the U.S. government announced a new intervention of income support. The aim of this stimulus was to cover the salaries or provide direct cash payments to people who had lost their jobs or could not work (e1). It also included payments to firms if explicitly linked to payroll or salaries. It was the most substantial stimulus package in U. S. history, and included payments of up to $1,200 for individuals or $2,400 for married couples. Parents also received $500 for each qualifying child. In spite of these generous fiscal steps, the results in Table 3 show a decline in the hospitality performance on the event date. This negative response is probably because this package of income support included payments to firms if explicitly linked to payroll or salaries but did not compensate the hospitality industry for its massive loss of revenues. In addition to the results reported above, additional findings for different time windows are available in the S1 Appendix. The results remained similar.

In the next subsection, we examine the impact of pandemic-driven uncertainty developed by Baker et al. on the price variations in the hospitality sector [46].

## Causality test results

Table 4 reports the Granger-causality test results about the causal relationship between uncertainty due to infectious disease and variations in the hospitality stock portfolio. We follow the literature and use the squared returns as a proxy for price variations (e.g., [59]). The table lists the F-statistics of the Granger-causality test. For robustness, we set the lag-length of the model using the Schwarz Bayesian information criterion and run the Granger procedure for five different order lags to gain insights into the dynamic relationship between the two variables.

The results, illustrated in Table 4, reports relatively large F-statistic values which indicates a major rejection of the null hypothesizes that uncertainty does not Granger-cause the variations

**Table 4. Granger causality test results (December 31, 2018-April, 30, 2020).**

|  | 1-Lag | 2-Lags | 3-Lags | 4-Lags | 5-lags |
|---|---|---|---|---|---|
| **Hospitality** |  |  |  |  |  |
| Uncertainty → Variation in Returns | 139.99*** | 52.92*** | 20.69*** | 13.93*** | 13.54*** |
| **Beer & Liquor** |  |  |  |  |  |
| Uncertainty → Variation in Returns | 64.59*** | 12.74*** | 4.67*** | 6.44*** | 6.56*** |
| **Apparel** |  |  |  |  |  |
| Uncertainty → Variation in Returns | 111.66*** | 28.49*** | 17.91*** | 17.71*** | 13.94*** |
| **Food** |  |  |  |  |  |
| Uncertainty → Variation in Returns | 41.41*** | 14.4*** | 8.19*** | 9.31*** | 7.31*** |
| **Fun** |  |  |  |  |  |
| Uncertainty → Variation in Returns | 34.54*** | 4.58** | 2.92** | 10.62*** | 17.35*** |
| **Consumer Goods** |  |  |  |  |  |
| Uncertainty → Variation in Returns | 40.65*** | 7.40*** | 9.14*** | 12.32*** | 13.71*** |
| **Personal Services** |  |  |  |  |  |
| Uncertainty → Variation in Returns | 71.55*** | 8.16*** | 9.11*** | 11.08*** | 18.22*** |
| **Candy& Soda** |  |  |  |  |  |
| Uncertainty → Variation in Returns | 134.23*** | 36.35*** | 15.72*** | 7.89*** | 11.06*** |
| **Transportation** |  |  |  |  |  |
| Uncertainty → Variation in Returns | 91.77*** | 18.89*** | 10.19*** | 7.89*** | 8.23*** |
| **Retail** |  |  |  |  |  |
| Uncertainty → Variation in Returns | 23.00*** | 6.84*** | 6.60*** | 9.43*** | 12.69*** |
| #Observations | 334 | 333 | 332 | 331 | 330 |

Notes: The table reports the results of the Granger causality test of the relationship between uncertainty due to infectious diseases and variations in returns in the hospitality and other related industries. The data cover December 31, 2018 to April 30, 2019. The results indicate that uncertainty due the COVID-19 pandemic is the driver of the variation in the stock prices in the hospitality sector. The values reported are the F-statistic values related to the Granger test. By Uncertainty → Variation in Returns, we mean that uncertainty does not Granger-cause the variations in returns. The lag order refers to the number of lags used in Eq (2).
"***," "**" and "*" denote statistical significance at the 1%, 5% and 10% levels, respectively.

in returns. In other words, the results suggest that the uncertainty driven by the COVID-19 pandemic is a strong driver of the variations taking place in the equities of hospitality firms. This result is in line with the view that the intensity of news and media coverage contributes to the variability of the stock market (e.g., [60]).

Table 4 illustrates the results of the test conducted on the data between December 31, 2018 and April 30, 2020. The results reported here are straightforward and indicate that this type of uncertainty drives price fluctuations in hospitality as well as other closely related industries. In line with a battery of studies documenting the link between the stage of the outbreak and equity returns (e.g., [10]), our results go one further step. They confirm that the most significant factor during pandemics, particularly for the hospitality industry, is uncertainty.

It is also worth noting that a possible source of the high levels of uncertainty might be the strong government interventions and the frequency of the interventions themselves. Pastor and Veronesi [23] presented a general-equilibrium model according to which changes in government policy create two types of uncertainty: impact uncertainty and political uncertainty. The former is the uncertain impact a particular policy will have on the profitability of firms. The latter is the general uncertainty resulting from changes in policy. In equilibrium, these types of uncertainty lead to increased volatility when the government changes its policy.

The results in Table 5 lend support to this contention. We examined the average values of uncertainty on the day of and the day following government intervention. For robustness, we

**Table 5. Uncertainty average on the day of and the day following intervention.**

| | Contemporaneous Model | | | One-lead Model | | |
|---|---|---|---|---|---|---|
| | $U_{it} = \beta_0 + \beta_1 \times Intervenstion_t + u_{it}$ | | | $U_{it+1} = \beta_0 + \beta_1 \times Intervenstion_t + v_{ti}$ | | |
| | **U1** | **U2** | **U3** | **U1** | **U2** | **U3** |
| Intercept | 18.98*** | 4.97*** | 0.03*** | 18.97*** | 5.05*** | 0.03*** |
| | (29.02) | (6.97) | (2.83) | (29.02) | (6.99) | (3.40) |
| Intervention | 29.48*** | 28.06*** | 0.36*** | 29.40*** | 26.04*** | 0.13** |
| | (8.16) | (7.12) | (6.17) | (8.13) | (6.53) | (2.03) |

Notes: The table reports the estimation results for two models linking intervention to uncertainty ($U_{it}$). The latter is captured using three different proxies (i = 1,2,3): U1 refers to the S&P 500 volatility index (VIX), U2 refers to the Bakers' et al. pandemic-driven uncertainty [46], and U3 is the squared daily returns of the hospitality sector. Intervention is a dummy variable that receives 1 if day t included government intervention and zero otherwise. The sample period is December 2018 to April 30, 2020, and the values in parentheses are the T-statistic values.

utilized three different proxies for uncertainty, and they include: the CBOE S&P 500 volatility index VIX, the pandemic-driven uncertainty developed by Baker et al. [46] and the squared returns of the hospitality sector. Overall, the picture emerged in Table 5 indicates that intervention days as well as the day following were associated with increased uncertainty. Therefore, one of our main conclusions is that governments should be transparent about their future actions and interventions regarding the hospitality industry.

Furthermore, decision makers and businesses operators in the hospitality industry should make their best efforts to increase their transparency and make their transactions with consumers as comfortable as possible. Reducing the degree of uncertainty is a key element in rebuilding the hospitality industry and making government actions efficient, with minimum financial and economic damage.

## Policy implications and recommendations

Our findings have important implications and suggestions for the hospitality industry managers, investors exposed to this industry, and policy makers at both the firm and state levels. We demonstrated that the impact of COVID-19, as well as government interventions, is not limited to the hospitality industry alone, but also affects other industries related to it. However, as a sector that relies on people's disposable income, the hospitality industry is particularly sensitive to economic upheaval. When the economic situation is uncertain, consumers typically tend to put off their consumption of hospitality in favor of more basic products. Furthermore, the sudden, widespread outbreak of COVID-19 caught everyone by surprise.

Governments groped in the dark in an effort to find ways of dealing with the situation. Their frequent changes in policy stoked the uncertainty surrounding the situation. One method that governments can use to reduce uncertainty and increase public confidence is transparency. Since time is crucial in managing a crisis, government policies should be announced publicly as soon as possible. To deal with the uncertainty related to the spread of the virus itself, governments must continuously provide as much information as possible about the evolution of the pandemic. For example, they can use campaigns, create special pandemic or crisis management websites, and provide contacts and information—all aimed at giving the public the maximum sense of control.

Regarding the second type of uncertainty, which relates to government policy, governments should be transparent about the types of interventions they plan to use, how they expect to implement them and their planned duration. Governments should publicize their plans for dealing with economic and non-economic issues, indicating how they will affect individuals,

investors, employees, employers and business owners. Moreover, since one of the main concerns is loss of revenue and the instability of the economy, the government should be proactive and set up vehicles such as a national savings fund to remove any ambiguity about the state's financial ability to manage the crisis. In addition, it should announce the availability of income aid, debt payment relief and fiscal spending that is immediate, widespread and generous. In addition, the public should be informed precisely how the government plans to finance this spending (imposing taxes, increasing deficits and debt ceilings, etc.). Transparent policies may mitigate the uncertainty in the short term and allay fears about the economy.

## Implications for investing

Our findings imply that investors, funds and portfolios managers should account for possible extreme events such as pandemics followed by governmental interventions, which may spillover to other industries and capital markets.

Investors can improve their investment tactics by paying more attention to the characteristics of the economic measures imposed to deal with infectious diseases and their resulting uncertainty, knowing now that they will have a negative impact on the performance of the hospitality industry. Hence, investors exposed to the hospitality industry should keep an eye on changes in Baker and colleagues' index of uncertainty due to infectious diseases. By doing so, they can safeguard the value of their investments using techniques such as short positions, derivatives, forwards or any financial instruments that have a weak or negative correlation with the performance of the hospitality and tourism sectors.

In the midst of COVID-19 and its economic implications, several opportunities may arise in other related industries, as evident in the unprecedented growth in their equity prices. According to AHLA's report, the hotel occupancy in the U.S. will increase to 52% in 2021, and to 61% in 2022, compared with 66% level in 2018–2019 (https://www.ahla.com/sites/default/files/2021_state_of_the_industry_0.pdf). In addition, the report projects that U.S. restaurant sales are about to increase to 11% in 2021 to $731.5 billion, but still far behind 2019's $864.3 billion. In this respect, future studies can examine the performance of hospitality and related industries before, during and after the COVID-19.

## Implications for policy making for the hospitality sector

While governments around the globe, including the U.S., have started to make interventions, recovery back to pre-crisis levels may not be immediate, but will take a longer time. During this time, owners and businesses operators in the hospitality industry can prepare for future activity and the post COVID-19 tourism environment. In addition, stakeholders should adjust their businesses for both the short and long terms and minimize the sensitivity of their businesses to the possibility of future government restrictions even after reopening the hospitality economy. They also have to consider innovations in their businesses in terms of social distancing and keeping a sanitized environment. Owners and hospitality service-related vendors such as airline companies, cruise companies and multinational hotel chains must cooperate with each other, and enforce strict protocols for the handling and preparing of food. Travel agencies can encourage people to travel by offering flexible rebooking options and free health insurance that covers COVID-19 during the trip. Fast COVID-19 tests before departure and after landing will help airline companies and airport authorities restore confidence in the safety of travel, which will encourage tourism and prevent infection in public areas.

For national policy makers, our results highlight the devastating response of the capital market to interventions, particularly closures. The main factor hurting the hospitality sector and other related industries is uncertainty. Indeed, at this point, there is still great uncertainty

about the rapidity of the spread of the disease and its lethality, whether a second wave of infection will occur, the time required to develop and distribute vaccines to the public, the real effectiveness and outcomes of social distancing and whether government policy responses and interventions will become permanent.

Therefore, governments should try to be as transparent as possible in devising a clear plan and definitive goals for the near and far future. They should do so by working with the other state economic authorities in order to minimize the negative impact of uncertainty. This approach is crucial because the cumulative evidence indicates that uncertainty reduces economic growth and firms postpone investment and hiring [46]. Furthermore, unclear, inconsistent government policy creates uncertainty that commands an equity risk premium, and may, in turn, affect the weighted average cost of capital, which ultimately affects firms' innovation activities (e.g., [61]).

Our results also underscore the adverse effect that closures have on the performance of the hospitality industry. Therefore, governments should be extremely transparent before enacting such regulations. They should also follow them with economic support to reduce not just the negative effect of the closures, but also the negative effect of the uncertainty about when the restrictions will be lifted. An important recommendation derived from our results is providing financial assistance alongside regulations that close workplaces. As our results hint, it is a key condition for providing significant relief from the detrimental impact of labor market closures. Since the labor market is an important component of the soundness and growth of the economy, especially in the hospitality industry, governments should consider mitigating the negative impact of closures with financial measures that support employees, employers and the economy. In this respect, it is important to consider the long-term implications of the COVID-19 crisis. Beyond their immediate impact, the measures put in place today will shape the future of hospitality. This outbreak is an opportunity for governments and businesses to develop new concepts of hospitality by reducing costs, utilizing green and clean energy and implementing new health protocols for safe travel. Governments should encourage the digital, low carbon, structural transitions needed to build a stronger, more sustainable and resilient hospitality economy.

Finally, scholars can use our findings to explain why the performance of the hospitality industry deviated so much from that of other service-oriented and economic sectors. They can incorporate the uncertainty resulting from infectious diseases and interventions into their pricing kernels.

To conclude, the massive fiscal stimulus adopted during the subprime crisis of 2008 showed that the intervention of government as well as other monetary and economic authorities was crucial in halting the financial deterioration of the finance sector and, consequently, the real economy (e.g., [61, 62]). Thus, the outcomes could have been even more destructive without the quick, massive, generous government measures. While these measures did not have an immediate positive effect because of the great uncertainty and public panic at that point, they might have had a lagged effect. Fig 2B depicts the gradual effect following the economic steps during the pandemic outbreak. Nevertheless, stimulus plans are controversial. Politicians must decide whether they want to stimulate the economy by increasing debt-financed spending, increasing balanced-budget spending financed through higher taxes in the future or providing debt-financed tax cuts.

## Summary and conclusions

This study explored the effects of interventions by the U.S. government on the market value and stock returns of the hospitality sector and industries closely related to it. We used an event

study methodology to explore the impact of four types of interventions: economic, health, closures and mixed types of interventions.

Closures had a consistently negative effect on the hospitality industry. The closing of public transportation (e5), domestic travel restrictions (e7) and stay at home requirements (e6) had a strong negative impact on the hospitality industry. Such interventions had a direct negative impact on the revenues of this industry.

The only intervention that had a significantly positive effect on the hospitality industry was the $100 billion COVID-19 aid package, signed by President Trump on the evening of March 18, 2020. Contrary to what we might expect, the economic interventions of income support (e1) and debt/contract relief (e2) had a significantly negative effect on the hospitality industry.

Possible explanations for these results might be the uncertainty originated in the unclear and inconsistent government policy itself. The latter is viewed in the public finance literature as a source of ambiguity with a direct effect on the stability of key macroeconomic variables. In addition, the hospitality sector expected more meaningful and substantial government support, especially, due to the expectation that this industry would have a late recovery. Similarly, the package provided payments to firms that were explicitly linked to payroll or salaries but did not compensate companies in the hospitality sector or closely related industries for their massive loss of revenue. Furthermore, the resulting uncertainty driven by the pandemic in terms of the spread of the pandemic, its lethality, time required to develop vaccines, unknown economic implications, etc., played a major role in the price fluctuations of hospitality equities. In periods of uncertainty, firms always invest less and hire fewer people. Similarly, people tend to save money rather than spend it. Accordingly, such unprecedented uncertainty was translated into a sharp decline in the value of hospitality firms.

These conclusions are relevant to both regulators and the leaders of the hospitality industry. Government policy is not just about choosing a deficit level, but also about influencing public expectations. Hence, leaders of the hospitality sector must prompt regulators to develop clear policies aimed at reducing economic uncertainty and persuade them that a gradual lifting of the closures will allow the hospitality industry to recover.

## Supporting information

**S1 Appendix. Additional results for different time windows.**
(DOCX)

## Author Contributions

**Conceptualization:** David Yechiam Aharon, Joseph Tzur.

**Data curation:** David Yechiam Aharon, Arie Jacobi.

**Formal analysis:** David Yechiam Aharon.

**Investigation:** David Yechiam Aharon, Arie Jacobi, Eli Cohen, Joseph Tzur, Mahmoud Qadan.

**Methodology:** David Yechiam Aharon, Mahmoud Qadan.

**Project administration:** David Yechiam Aharon.

**Resources:** Mahmoud Qadan.

**Software:** David Yechiam Aharon, Mahmoud Qadan.

**Supervision:** David Yechiam Aharon.

**Validation:** Arie Jacobi, Eli Cohen, Joseph Tzur.

**Visualization:** Arie Jacobi, Eli Cohen, Joseph Tzur.

**Writing – original draft:** David Yechiam Aharon, Arie Jacobi, Eli Cohen, Joseph Tzur.

**Writing – review & editing:** David Yechiam Aharon, Arie Jacobi, Eli Cohen, Joseph Tzur, Mahmoud Qadan.

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
