## [Decision Letter · Decision Letter 0]

19 Apr 2021

PONE-D-21-07279

COVID-19, Government Measures and Hospitality Industry Performance: An Event Study Approach

PLOS ONE

Dear Dr. Aharon,

Thank you for submitting your manuscript to PLOS ONE. After careful consideration, we feel that it has merit but does not fully meet PLOS ONE’s publication criteria as it currently stands. Therefore, we invite you to submit a revised version of the manuscript that addresses the points raised during the review process.

The paper shows merit, but several shortcomings should be fixed. For instance, the contribution of the manuscript should be strongly argued. As well, there is required an extended discussion of the existing literature regarding the influence of COVID-19 impact on hospitality industry, alongside other clarifications (e.g., hospitality sector vs tourism and hospitality).

We look forward to receiving your revised manuscript.

Kind regards,

Stefan Cristian Gherghina, PhD. Habil.

Academic Editor

PLOS ONE

Journal Requirements:

Please ensure that you include a title page within your main document. We do appreciate that you have a title page document uploaded as a separate file, however, as per our author guidelines (http://journals.plos.org/plosone/s/submission-guidelines#loc-title-page) we do require this to be part of the manuscript file itself and not uploaded separately.

Please include captions for your Supporting Information files at the end of your manuscript, and update any in-text citations to match accordingly. Please see our Supporting Information guidelines for more information: http://journals.plos.org/plosone/s/supporting-information.

Reviewers' comments:

Reviewer's Responses to Questions

**Comments to the Author**

1. Is the manuscript technically sound, and do the data support the conclusions?

Reviewer #1: Yes

Reviewer #2: Yes

Reviewer #3: Yes

2. Has the statistical analysis been performed appropriately and rigorously? 

Reviewer #1: Yes

Reviewer #2: Yes

Reviewer #3: Yes

3. Have the authors made all data underlying the findings in their manuscript fully available?

Reviewer #1: Yes

Reviewer #2: Yes

Reviewer #3: Yes

4. Is the manuscript presented in an intelligible fashion and written in standard English?

Reviewer #1: Yes

Reviewer #2: Yes

Reviewer #3: Yes

5. Review Comments to the Author

Reviewer #1: I commend the authors on providing timely analysis of policy interventions on hospitality stocks. The paper was very clear and flowed very nicely. My only suggesting would be to have some discussion related to using market prices as a proxy for the overall health/state of the hospitality industry. Its a solid measure, but implies some limitations worth mentioning in my opinion.

Reviewer #2: 1. The introduction talk about COVID-19 impact and it is need to express health impact of COVID-19 in terms and people and died and people detected. It must use the information of WHO (https://covid19.who.int)

2. In Literature review it must include goverments interventions to control fake news which affect usual commerce in a country (https://doi.org/10.4269/ajtmh.20-0536).

3. One of the first evaluation of impact in hospitality workers by COVID-19 must be included (https://doi.org/10.1016/j.ijhm.2021.102935)

4. Explain in which country was developed the analysis by Kizys et al. (2020), Huang et al. (2020), Adda (2016), Pennathir (2014).

Reviewer #3: I do not like the way in which Authors explain their contribution. This is relatively obvious that they work contributes to the COVID-19, Hospitality industry and asset pricing or government intervention literature.

What I expect from the publication at this level is to clearly state HOW they contribute. What I mean more specifically is that Authors should say: we contribute to the research dealing with impact of COVID-19 (…) on hospitality and tourism sector BY …. [PLEASE SPECIFY]. Please specify in similar vein you contribution to the remainder streams of the literature mentioned in the contribution. I suggest to combine this with the results you discuss briefly in the introduction.

The literature review is missing the context of hospitality & tourism industry perspective. Please consider re-naming section 2.2. (by adding the reference to the sector in focus). Apart from one study cited in this section (Chen et al. 2007) all other references are related to the COVID-19 and stock performance. I recommend to add the literature on overall COVID-19 impact on hospitality industry. This will help you (1) to explain more in depth of what you observe in stock reurns and (2) highlight an important contribution of your work in terms of hospitality & travel sector (they are very much concerned about tourism arrivals, far less about the financially-oriented perspective, such as link to economy and strock market). You will find plety of sources in the thematic journals in the field, such as Tourism Management or International Journal of Hospitality Management

I recommend also to carefully revise the text to use unified reference to the sector in focus. You use the term „hospitality sector” interchangeably with „tourism and hospitality”. At the same time there are sentences that pose hospitality as a separate sector from travel and leisure (e.g. in the introduction „travel, leisure, hospitality and tourism industries”). Sometimes you even report reserach or data from „travel and tourism” sector. In my opinion you should inform a reader how you understand „hospitality industry” and whether you adopt a very broad defininion placing under this umbrella the whole travel and leisure sector. Given the sample construction (and the SIC codes you listed) – this need to be clearly stated in this paper.

Good luck!

6. PLOS authors have the option to publish the peer review history of their article (what does this mean?). If published, this will include your full peer review and any attached files.

Reviewer #1: No

Reviewer #2: No

Reviewer #3: No

---

## [Author Response · Author response to Decision Letter 0]

28 May 2021

Reply to the Editor 

Manuscript No.: PONE-D-21-07279

Title: COVID-19, Government Measures and Hospitality Industry Performance 

We would like to thank you for your constructive comments. In the enclosed revised version, we tried our best to answer the points raised and incorporate them into the revised manuscript. We definitely believe that your comments improved the clarity, readability and quality of the paper. They are addressed here in the order they appear in your report followed by our reply.

Editor comments

Thank you for submitting your manuscript to PLOS ONE. After careful consideration, we feel that it has merit but does not fully meet PLOS ONE’s publication criteria as it currently stands. Therefore, we invite you to submit a revised version of the manuscript that addresses the points raised during the review process.

The paper shows merit, but several shortcomings should be fixed. For instance, the contribution of the manuscript should be strongly argued. As well, there is required an extended discussion of the existing literature regarding the influence of COVID-19 impact on hospitality industry, alongside other clarifications (e.g., hospitality sector vs tourism and hospitality).

Answer: Thanks for your support and positive spirit. In line with your comments, we did our best to address all of the points raised by the reviewers. Regarding the two specific points you noted above, please see our response to the first and second comments of reviewer #3 and the changes we made as a result, which are listed point by point. In a nutshell, we highlighted our contribution in a stronger way, and extended the literature review with the additional material suggested. 

Answer: Following these points:

• We included a rebuttal letter as a separate file, called 'Response to Reviewers'.

• We uploaded a separate marked-up copy of our manuscript labeled 'Revised Manuscript with Track Changes'. The marked-up manuscript uses “track changes” to meet both the technical requirements of the journal and the concerns raised by the referees.

• We uploaded a separate unmarked copy of our manuscript labeled 'Manuscript'.

Journal Requirements:

and

Answer: Thanks for drawing our attention to these requirements. We adjusted the entire manuscript to meet the general guidelines and to ensure that the file names correspond to the journal's requirements.

Answer: We apologize for this. The title page is now located within the main document, at its beginning. 

Answer: Thanks. Done. Now we include a caption for the online appendix tables at the end of our manuscript (S1 Appendix. Additional results for Different time windows). At the end of the empirical findings section, we refer the reader to the supporting file.

"In addition to the results reported above, additional findings for different time windows are available in the S1 Appendix. The results remained similar."

 

Reply to Referee#1 

Manuscript No.: PONE-D-21-07279

Title: COVID-19, Government Measures and Hospitality Industry Performance 

We would like to thank you for your constructive comments. In the enclosed revised version, we tried our best to answer the points raised and incorporate them into the revised manuscript. We definitely believe that your comments improved the clarity, readability and quality of the paper. They are addressed here in the order they appear in your report followed by our reply.

I commend the authors on providing timely analysis of policy interventions on hospitality stocks. The paper was very clear and flowed very nicely. My only suggesting would be to have some discussion related to using market prices as a proxy for the overall health/state of the hospitality industry. Its a solid measure, but implies some limitations worth mentioning in my opinion.

Answer: We very much appreciate your positive feedback and support – thank you. Indeed, the use of market prices of firms is a solid measure and is also used in the tourism and hospitality industry as an acceptable and reliable proxy capable of reflecting the overall state of the hospitality industry. This perception is motivated by the classic paradigm in finance maintaining that security prices reflect all available information in the market place. In line with your comment, some limitations of this paradigm may refer to pricing anomalies originating in herd behavior or other behavioral biases. However, the empirical and theoretical literature regard these deviations as temporary. In the data section we added a short discussion justifying the use of the market prices of tourism and hospitality firms to reflect the state of the hospitality industry. The addition is as follows:

"… We use market prices as a proxy for the overall state of the hospitality industry as well as for the other related sectors. This approach might have limitations, albeit temporary ones, which stem from behavioral biases. Nevertheless, using market prices is still a prevalent method that reflects the present value and state of securities."

Finally, we would like to thank you again for your comments and suggestions.

Sincerely, 

The authors

 

Reply to Referee#2 

Manuscript No.: PONE-D-21-07279

Title: COVID-19, Government Measures and Hospitality Industry Performance

We would like to thank you for your constructive comments. In the enclosed revised version, we tried our best to answer the points raised and incorporate them into the revised manuscript. We definitely believe that your comments improved the clarity, readability and quality of the paper. They are addressed here in the order they appear in your report followed by our reply.

1. The introduction talk about COVID-19 impact and it is need to express health impact of COVID-19 in terms and people and died and people detected. It must use the information of WHO (https://covid19.who.int)

Answer: Thanks for your feedback. We added a discussion about this in the introduction section using data from the WHO. The addition is located in the 3rd paragraph in the introduction section.

"According to the World Health Organization (WHO), the COVID-19 pandemic was first reported in Wuhan, China on December 31, 2019. The pandemic spread quickly all over Asia, leaving behind it health and economic crises. On March 2, 2020, COVID-19 was first reported in the US and 10 days later Europe became the epicenter of the pandemic, both leading to even worse health and economic catastrophes. On June 1, 2020, there were over 6 million confirmed cases, and more than 370,000 confirmed deaths worldwide. Remarkably, as of May 20, 2021, the WHO reported that there had been 166,346,635 confirmed cases of COVID-19, including 3,449,117 deaths." 

2. In Literature review it must include governments' interventions to control fake news which affect usual commerce in a country (https://doi.org/10.4269/ajtmh.20-0536).

Answer: Thanks for your comment, and guiding us to this excellent source. We updated the literature review to include this valuable study. Specifically, for this citation we added the following review:

"The COVID-19 pandemic has generated an unprecedented level of public fear, which in some cases even intensified due to the spreading of incorrect information or fake news mainly on social media. Some governments, such as in Peru, were successful in combatting this fake news by imposing prison sentences on those who created and shared fake news (Alvarez-Risco et al. [33])"

- Alvarez-Risco, A., Mejia, C. R., Delgado-Zegarra, J., Del-Aguila-Arcentales, S., Arce-Esquivel, A. A., Valladares-Garrido, M. J.,Rosas del Portal, M.; Villegas, L.F., Curioso, W.H., Sekar, M. C, & Yáñez, J. A. (2020). The Peru approach against the COVID-19 infodemic: insights and strategies. The American Journal of Tropical Medicine and Hygiene, 103(2), 583-586.‏

3. One of the first evaluation of impact in hospitality workers by COVID-19 must be included (https://doi.org/10.1016/j.ijhm.2021.102935)

Answer: Thanks for your feedback. In line with the previous comment, we added this additional source as part of our literature review.

Specifically, we added the following mention of this paper in our literature review section:

"Yan et al. [36] used data from 211 hospitality workers in 76 hotels in Peru to examine the effects of perceptions about COVID-19 risks on the likelihood of experiencing depressive symptoms. They found that job satisfaction weakens the relationship between hospitality workers’ COVID-19 risk perceptions and their likelihood of depressive symptoms. On the other hand, they also found that the number of children the workers have aggravates this relationship."

- Yan, J., Kim, S., Zhang, S. X., Foo, M. D., Alvarez-Risco, A., Del-Aguila-Arcentales, S., & Yáñez, J. A. (2021). Hospitality workers’ COVID-19 risk perception and depression: A contingent model based on transactional theory of stress model. International Journal of Hospitality Management, 95, 102935.‏

4. Explain in which country was developed the analysis by Kizys et al. (2020), Huang et al. (2020), Adda (2016), Pennathir (2014).

Answer: Thanks for your comment. We added this important information the discussion in the literature review. Specifically, Kizys et al. (2020) used 72 stock market indices from both developed and emerging economies, Adda (2016) employed data from France, while both Huang et al. (2020) and Pennathir (2014) relied on US data.

Finally, we would like to thank you again for your comments and suggestions.

Sincerely, 

The authors 

Reply to Referee#3 

Manuscript No.: PONE-D-21-07279

Title: COVID-19, Government Measures and Hospitality Industry Performance: An Event Study Approach  

PLOS ONE

We would like to thank you for your constructive comments. In the enclosed revised version, we tried our best to answer the points raised and incorporate them into the revised manuscript. We definitely believe that your comments improved the clarity, readability and quality of the paper. They are addressed here in the order they appear in your report followed by our reply.

I do not like the way in which Authors explain their contribution. This is relatively obvious that they work contributes to the COVID-19, Hospitality industry and asset pricing or government intervention literature.

What I expect from the publication at this level is to clearly state HOW they contribute. What I mean more specifically is that Authors should say: we contribute to the research dealing with impact of COVID-19 (…) on hospitality and tourism sector BY …. [PLEASE SPECIFY]. Please specify in similar vein you contribution to the remainder streams of the literature mentioned in the contribution. I suggest to combine this with the results you discuss briefly in the introduction.

Answer: Thank you for raising this important point. Following your comment, we reorganized the introduction to sharpen our contribution in the spirit suggested in your comment. We combined the points with the results presented in the Introduction. The addition is as follows:

"We contribute to the literature in three areas. First, we add to the growing body of research dealing with the impact of epidemics and crises on asset pricing (e.g., [2-15]) by examining the immediate and short-term effect of the COVID-19 outbreak on the price evolution of stocks in the hospitality and tourism sectors using the event study method. Second, we contribute to the literature dealing with the impact of government interventions and their reflection in asset prices during times of crisis (e.g., [11, 16-22]) by sorting the intervention into economic (e.g., income support, debt contract relief) and non-economic (e.g., travel restrictions, school closings) measures and exploring their consequences. Overall, the results show that both economic and non-economic interventions imposed on the public can affect the prices of tourism and hospitality stocks. The magnitude of the short-term negative effect increases with the timeline of the evolution of the epidemic and the increased level of uncertainty. 

Finally, we contribute to the literature dealing with changes in government policy and uncertainty in the stock market (e.g., [23]), by exploring the relationship between uncertainty due to infectious disease and variations in the hospitality stock returns. In addition, we examine how uncertainty reacts to government intervention. The results highlight that the hospitality business is very sensitive to economic uncertainty. When faced with adverse economic conditions, consumers typically tend to postpone using disposable income for travel and tourism in favor of more basic needs [24]." 

The literature review is missing the context of hospitality & tourism industry perspective. Please consider re-naming section 2.2. (By adding the reference to the sector in focus). Apart from one study cited in this section (Chen et al. 2007) all other references are related to the COVID-19 and stock performance. I recommend to add the literature on overall COVID-19 impact on hospitality industry. This will help you (1) to explain more in depth of what you observe in stock returns and (2) highlight an important contribution of your work in terms of hospitality & travel sector (they are very much concerned about tourism arrivals, far less about the financially-oriented perspective, such as link to economy and stock market). You will find plenty of sources in the thematic journals in the field, such as Tourism Management or International Journal of Hospitality Management

Answer: Thanks for this insightful point. Following your comment, we renamed subsection 2.2. It is now entitled: "COVID-19’s impact on the hospitality industry". This subsection provides a better view of the overall impact of COVID-19 on the hospitality industry. Thanks to your comment we were able to connect the two strands of the literature to the expected results of government interventions in the hospitality industry. Formally, we added the following:

COVID-19’s impact on the hospitality industry

" According to the Economic Impact Report by the World Travel and Tourism Council [30], prior to the pandemic, the travel and tourism sectors, both directly and indirectly, accounted for 1 in 4 of all new jobs created around the world, 10.6% of all jobs (334 million), and 10.4% of global GDP (US$9.2 trillion). In 2020, 62 million jobs were lost, representing a drop of 18.5% (62/334). The threat of job losses is continuing as many jobs are currently supported by government retention schemes and reduced hours, which without a full recovery in this sector, could be lost.

The employees in the hotel business were suddenly unemployed and their job security plummeted. Jung et al. [31] studied the perceptions of five-star hotel employees in Seoul that provide comprehensive services such as restaurants and gyms, with 200 or more bedrooms (22 hotels that have, on average, 500 employees in food and beverage services). The study showed that perceptions of job insecurity had negative effects on the engagement of deluxe hotel employees. These results are telling because employees’ job engagement can reduce their turnover intentions. Furthermore, the job insecurity caused by COVID-19 had a stronger negative influence on Generation Y than Generation X in reducing job engagement. 

Recently, Hao et al. [32] reviewed the overall impact of the pandemic on China’s hospitality industry - the country where the health crisis began. The industry witnessed a sharp decline in hotel occupancy rates and a loss of over U.S.$9 billion in revenue. About 74% of the hotels in China were closed in January and February 2020 for an average period of 27 days. Furthermore, from January 14 to 28, the occupancy of the hotels dropped from around 70% to 8% and remained under 10% in the following 28 days. As a result, the hotel and tourism industry reduced their number of employees, leading to a significant drop in cash flow and revenue. 

The COVID-19 pandemic has generated an unprecedented level of public fear, which in some cases even intensified due to the spreading of incorrect information or fake news mainly on social media. Some governments, such as in Peru, were successful in combatting this fake news by imposing prison sentences on those who created and shared fake news (Alvarez-Risco et al. [33]).

Zheng et al. [34] studied the phenomenon of “travel fear” in China. They reported that perceptions about the severity of the threat and the susceptibility to it can cause “travel fear,” which leads to protective behaviors with regard to travel decisions. Furthermore, “travel fear” can evoke different strategies that increase people’s psychological resilience and adoption of cautious travel behaviors. Villace-Molinero et al. [35] explored perceptions about travel risks during the pandemic and proposed measures to improve traveler confidence based on the issue-attention cycle. Based on a survey conducted in 46 countries and a qualitative study in which 28 international hospitality experts were interviewed, the authors concluded that in a pandemic scenario, confidence in communications from the local government about personal safety and security are the main factors people consider when making travel decisions. Finally, Yan et al. [36] used data from 211 hospitality workers in 76 hotels in Peru to examine the effects of perceptions about COVID-19 risks on the likelihood of experiencing depressive symptoms. They found that job satisfaction weakens the relationship between hospitality workers’ COVID-19 risk perceptions and their likelihood of depressive symptoms. On the other hand, they also found that the number of children the workers have aggravates this relationship.

To summarize, these studies show that in addition to the negative effect that government interventions usually have on financial markets, COVID-19 also had various detrimental effects on the hospitality industry. These effects were evident in the labor force, the willingness to travel, and the spreading of fake news. Therefore, combining these pieces of evidence, we might expect that the impact of government interventions on the hospitality industry would also be negative."

References

World Travel and Tourism Council (WTTC) (2021). Economic Impact Reports. https://wttc.org/Research/Economic-Impact (accessed on April 22, 2021). 

Jung, H.S., Jung, Y.S. and Yoon, H.H. (2021). COVID-19: The effects of job insecurity on the job engagement and turnover intent of deluxe hotel employees and the moderating role of generational characteristics. International Journal of Hospitality Management 92 (2021), 102703.

Hao, F., Xiao, Q. and Chon, K. (2020). COVID-19 and China’s Hotel Industry: Impacts, a Disaster Management Framework, and Post-Pandemic Agenda. International Journal of Hospitality Management, Vol. 90, September 102636.

Alvarez-Risco, A., Mejia, C.R., Delgado-Zegarra, J. Del-Aguila-Arcentales, S. Arce-Esquivel, A.A., Valladares-Garrido, M.J., Rosas del Portal, M., Villegas, L.F., Curioso, W.H., Sekar,M.C. and Yañez, J.A. (2020). Perspective Piece. The Peru Approach against the COVID-19 Infodemic: Insights and Strategies. American Journal of Tropical Medicine and Hygiene, 103(2), 583–586. doi:10.4269/ajtmh.20-0536.

Zheng, D. Luo, Q. and Ritchie, B.W. (2021).Afraid to travel after COVID-19? Self-protection, coping and resilience against pandemic ‘travel fear’. Tourism Management 83, 104261.

Villace-Molinero, T., Fernandez-Munos, J.J., Orea-Giner, A. and Fuentes-Moraleda, L. (2021). Understanding the new post-COVID-19 risk scenario: Outlooks and challenges for a new era of tourism. Tourism Management 86 (2021) 104324.

I recommend also to carefully revise the text to use unified reference to the sector in focus. You use the term „hospitality sector” interchangeably with „tourism and hospitality”. At the same time there are sentences that pose hospitality as a separate sector from travel and leisure (e.g. in the introduction „travel, leisure, hospitality and tourism industries”). Sometimes you even report research or data from „travel and tourism” sector. In my opinion you should inform a reader how you understand „hospitality industry” and whether you adopt a very broad definition placing under this umbrella the whole travel and leisure sector. Given the sample construction (and the SIC codes you listed) – this need to be clearly stated in this paper.

Good luck!

Answer: Thanks for your sharp eyes. Following your comment, we followed your suggestion and used the hospitality and tourism sectors as an umbrella covering all related sectors. We added a clarification in this spirit in the data section.

"Our sample consists of the daily log returns of stock portfolios consisting of firms operating in the hospitality industry in the following COMPUSTAT SIC codes: Retail‒eating places (5800‒5819), Restaurants, hotels, motels (5820‒5829), Eating and drinking places (5890‒5899), Hotels, other lodging places (7000‒7000), Hotels and motels (7010‒7019), Membership hotels and lodging (7040‒7049) and Services – linen (7213‒7213). We refer to these related industries collectively as the hospitality and tourism industry." 

Finally, we would like to thank you again for your comments and suggestions.

Sincerely, 

The authors

---

## [Decision Letter · Decision Letter 1]

11 Jun 2021

PONE-D-21-07279R1

COVID-19, Government Measures and Hospitality Industry Performance

PLOS ONE

Dear Dr. Aharon,

Thank you for submitting your manuscript to PLOS ONE. After careful consideration, we feel that it has merit but does not fully meet PLOS ONE’s publication criteria as it currently stands. Therefore, we invite you to submit a revised version of the manuscript that addresses the points raised during the review process.

The submission improved in a proper manner, but there are required further revisions towards literature review and policy/practical implications.

We look forward to receiving your revised manuscript.

Kind regards,

Stefan Cristian Gherghina, PhD. Habil.

Academic Editor

PLOS ONE

Journal Requirements:

Reviewers' comments:

Reviewer's Responses to Questions

**Comments to the Author**

1. If the authors have adequately addressed your comments raised in a previous round of review and you feel that this manuscript is now acceptable for publication, you may indicate that here to bypass the “Comments to the Author” section, enter your conflict of interest statement in the “Confidential to Editor” section, and submit your "Accept" recommendation.

Reviewer #1: (No Response)

Reviewer #2: (No Response)

Reviewer #3: (No Response)

2. Is the manuscript technically sound, and do the data support the conclusions?

Reviewer #1: Partly

Reviewer #2: Yes

Reviewer #3: Yes

3. Has the statistical analysis been performed appropriately and rigorously? 

Reviewer #1: Yes

Reviewer #2: Yes

Reviewer #3: Yes

4. Have the authors made all data underlying the findings in their manuscript fully available?

Reviewer #1: Yes

Reviewer #2: Yes

Reviewer #3: Yes

5. Is the manuscript presented in an intelligible fashion and written in standard English?

Reviewer #1: Yes

Reviewer #2: Yes

Reviewer #3: Yes

6. Review Comments to the Author

Reviewer #1: I appreciate the authors' effort to incorporate changes based on the initial reviews. I do however think that some clarity was lost in effort to broaden the hospitality sector impacts from COVID-19. It became less clear what challenges were brought on by COVID-19 itself and which were directly related to particular governmental interventions. The main finding seems to be that governments should reduce uncertainty, but I did not walk away from the paper knowing precisely how that might have been done, or could be done in the future, given that there was such a tremendous amount of uncertainty related to a novel virus circulating. The authors also included information about the negative effects of the impacts of lay-offs and employment instability on hospitality employees in the lit review, but then does not discuss this in the discussion or findings alongside of their results. Instead, the authors directly suggest that cutting costs (does that not mean labor?) is a recommendation to the industry to perform better post pandemic. As I initially suggested, market prices leave out these types of non market human/environmental impacts. I think expanding your lit review to discuss some of these impacts requires a more thorough discussion alongside of them and this discussion should inform your recommendations. I also think your recommendations could be more specific, particularly, if government interventions somehow directly increased uncertainty, what would uncertainty reduction policies have looked like or could look like in the future. Also, it isn't your fault necessarily, its a peril of trying to do timely research in rapidly fluctuated time period, but now we are seeing a rapid recovery in tourism and hospitality, particularly so in domestic hospitality. Can anything be gathered through an understanding of what sub-sectors or industry types are flourishing in this environment? Is it much of the same as the pre-pandemic or has their been any shift?

Reviewer #2: I have reviewed the article and it fullfill all the suggestions sent previously. It is ready for publication.

Reviewer #3: What I meant in my comment was (a) specification what do you understand by hospitality industry [and it was done], and (b) correction of the reference to hospitality industry in the whole body of the text [this still needs to be revised]. Please consider the examples of the mix of terms you use through the text (and this is only from a part of the text!)

Title: hospitality industry

Abstract, line 53 „hospitality and tourism industries”

Then, line 55 “tourism and hospitality”

87 “hospitality and tourism sectors”

93 “tourism and hospitality stocks” but in line 98 “hospitality stock returns”

And then in the empirical section: 250 “firms operating in the hospitality industry”

I have also some reservations to the literature review. Although it was expanded (to meet the review comments), still the literature review in the section “COVID-19 impact on the hospitality industry” (line 195 and the following) could be much better.

My first suggestion is to remain more concerned on hospitality industry itself (as this is the central point of your work). This section begins with the reference to statistics by WTTC, which is fine, but provides some general information on COVID-19 impact on TRAVEL AND TOURISM sector. Please try to add some statistics & data that refer directly to hospitality industry itself (e.g. https://hoteltechreport.com/news/tourism-industry-statistics#hotels, empirical works or industry reports)

My second suggestion is the revision of the empirical works cited in this section. Overall, it is good that on the literature review you provide examples directly tied to hospitality industry (not travel & tourism). However, still, given the extent of the current debate on COVID-19 impacts on hospitality industry, the evidence you provide in literature review seems scarce. Moreover, it is not convincing why these specific works have been revised in the literature review. As a reader, I am missing a clear “story behind”. What I mean more specifically, you refer to selected works, of very different scope, but there is no clear map that could guide us through these exemplary works cited within. You summarize this by stating (line 243): detrimental effects on the hospitality industry. But for me the employees job security (line 203), fake news (line 224) or depressive symptoms of workers (line 236) seem far away from the main problem (stock returns). As your study highlights the context of performance, you should search for the examples of prior works that conform to the performance-related perspective (loss of income, additional costs, overall – what increases the anxieties of investors). The examples of Hao et al. (line 212) or Zheng et al (line 225) are the right direction. Please try to add some more works of this kind, from a business and performance oriented perspective, to make your discussion more substantial. I recommend also to address the works that have revised the impacts of prior pandemic (SARS in 2002-2003 in particular), as these were the first to communicate the possible extent of the pandemic impact on hospitality industry (given the customer outflow and loss of income).

Good luck!

7. PLOS authors have the option to publish the peer review history of their article (what does this mean?). If published, this will include your full peer review and any attached files.

Reviewer #1: No

Reviewer #2: No

Reviewer #3: No

---

## [Author Response · Author response to Decision Letter 1]

11 Jul 2021

Dear Editor/Reviewers

Please see the detailed response letter, which we uploaded separately. The response letter addresses each of the comments and suggestions.

Thanks,

The Authors

---

## [Decision Letter · Decision Letter 2]

26 Jul 2021

COVID-19, Government Measures and Hospitality Industry Performance

PONE-D-21-07279R2

Dear Dr. Aharon,

We’re pleased to inform you that your manuscript has been judged scientifically suitable for publication and will be formally accepted for publication once it meets all outstanding technical requirements.

Kind regards,

Stefan Cristian Gherghina, PhD. Habil.

Academic Editor

PLOS ONE

Additional Editor Comments (optional):

Reviewers' comments:

Reviewer's Responses to Questions

**Comments to the Author**

1. If the authors have adequately addressed your comments raised in a previous round of review and you feel that this manuscript is now acceptable for publication, you may indicate that here to bypass the “Comments to the Author” section, enter your conflict of interest statement in the “Confidential to Editor” section, and submit your "Accept" recommendation.

Reviewer #1: All comments have been addressed

Reviewer #2: All comments have been addressed

Reviewer #3: (No Response)

2. Is the manuscript technically sound, and do the data support the conclusions?

Reviewer #1: Yes

Reviewer #2: Yes

Reviewer #3: Yes

3. Has the statistical analysis been performed appropriately and rigorously? 

Reviewer #1: Yes

Reviewer #2: Yes

Reviewer #3: Yes

4. Have the authors made all data underlying the findings in their manuscript fully available?

Reviewer #1: Yes

Reviewer #2: Yes

Reviewer #3: Yes

5. Is the manuscript presented in an intelligible fashion and written in standard English?

Reviewer #1: Yes

Reviewer #2: Yes

Reviewer #3: Yes

6. Review Comments to the Author

Reviewer #1: I think this manuscript is greatly improved. I commend the authors' for their effort and recommend for publication.

Reviewer #2: Congratulations. The article have been improved ad have enough academic support to be ready for publication.

Reviewer #3: Dear Authors,

in the resubmitted manusript my suggestions have been included.

This is an interesting study, congratulations!

7. PLOS authors have the option to publish the peer review history of their article (what does this mean?). If published, this will include your full peer review and any attached files.

Reviewer #1: **Yes: **Leon Mach

Reviewer #2: No

Reviewer #3: No

---

## [Editor Report · Acceptance letter]

27 Jul 2021

PONE-D-21-07279R2 

COVID-19, government measures and hospitality industry performance 

Dear Dr. Aharon:

I'm pleased to inform you that your manuscript has been deemed suitable for publication in PLOS ONE. Congratulations! Your manuscript is now with our production department. 

Kind regards, 

on behalf of

Dr. Stefan Cristian Gherghina 

Academic Editor

PLOS ONE